Effects of four different times of day on various aspects of maximal short-term physical performance in young soccer players

Bougrine Houda 1 2
Nasser Nidhal 1 3
Gandouzi Imed 4
Ceylan Halil ibrahim halil.ceylan@atauni.edu.tr halil.ibrahimceylan60@gmail.com 5
Bouazizi Majdi 6
Paillard Thierry 2
Dergaa Ismail 7
Stefanica Valentina valentina.stefanica@upb.ro 8
Abderrahman Abderraouf Ben 7
1 Physical Activity Research Unit, Sport and Health (UR18JS01), National Observatory of Sports , Tunis , Tunisia
2 Department of Sport Sciences, E2S UPPA, MEPS Laboratory, Université de Pau et des Pays de l’Adour , Tarbes , France
3 High Institute of Sport and Physical Education, University of Sfax , Sfax , Tunisia
4 Molecular Basis of Human Pathology Laboratory, Faculty of Medicine of Sfax , Tunis , Tunisia
5 Department of Physical Education of Sports Teaching, Faculty of Sports Sciences, Atatürk University , Erzurum , Turkey
6 High Institute of Sport and Physical Education, University of Gafsa , Tunis , Tunisia
7 Higher Institute of Sport and Physical Education of Ksar Said, University of Manouba , Tunis , Tunisia
8 Department of Physical Education and Sport, Faculty of Sciences, Physical Education and Informatics, Pitesti University Center, University Politehnica of Bucharest , Pitesti , Arges , Romania
Li Yumeng
Electronic publication date: 2025 Oct 22
Publication date: 2025
Volume: 13
Electronic Location ID: e20065
Received 2025 Feb 26; Accepted 2025 Aug 20
Copyright: ©2025 Bougrine et al.
Copyright year: 2025
Copyright holder: Bougrine et al.
License: This is an open access article distributed under the terms of the Creative Commons Attribution License, which permits unrestricted use, distribution, reproduction and adaptation in any medium and for any purpose provided that it is properly attributed. For attribution, the original author(s), title, publication source (PeerJ) and either DOI or URL of the article must be cited.
License URL: https://creativecommons.org/licenses/by/4.0/

Keywords: Time of day, Diurnal variation, Maximal performance, Adolescent, Agility, Sprint

Funding: The authors received no funding for this work.

==============================
The time of day (TOD) has a significant influence on physical performance through circadian rhythms, which regulate body temperature, muscle function, and hormone levels. While extensively studied in adults, the impact of TOD on adolescent athletes remains underexplored. This study aimed to investigate the effects of TOD on maximal short-term physical performance in adolescent soccer players, to guide evidence-based decisions regarding the structuring of training and competition schedules. Nineteen male adolescent soccer players (mean age: 14.58 ± 0.7 years) participated in a randomized trial, performing at four TOD sessions (08:00 h, 11:00 h, 15:00 h, and 18:00 h) with recovery periods of at least 48 hours between sessions. Performance metrics included oral temperature (OT), medicine ball throw (MBT), 5-meter sprint (5m-T), 20-meter sprint (20m-T), Illinois Agility Test (IAT), and Illinois Agility Test with Ball (IAT-B). Results revealed significant TOD effects. OT increased notably from 08:00 to 18:00 (p < 0.001). Physical performance metrics (MBT, 5m-T, 20m-T, IAT, and IAT-B) improved significantly at 11:00, 15:00, and 18:00 (all p < 0.05) relative to 08:00. The 18:00 session consistently produced the highest performance levels. At the same time, no significant differences were observed between 11:00 and 15:00 (all p > 0.05). Our results suggest that time of day significantly influences short-term maximal physical performance in adolescent male soccer players, with peak performance levels consistently observed in the late afternoon (18:00 h). This appears to reflect diurnal variations in core temperature, hormonal activity, and neuromuscular readiness, which likely enhance sprint speed, agility, and explosive power at this time. These findings indicate that coaches and youth sport practitioners should consider scheduling high-intensity training sessions and performance assessments in the late afternoon to optimize physiological conditions for training and evaluation. Aligning training and competition timing with circadian rhythms may therefore support improved athletic output and more accurate performance monitoring in youth soccer settings.

Introduction

The periodic responses of organisms to their environmental stimuli, such as the day-night cycle, seasonal variations, and moon-influenced tidal motions, have been identified as biological rhythms (Allebrandt et al., 2014; Bayer et al., 2023; Pradhan et al., 2024). Circadian rhythms influence essential human physiological and behavioral processes, which play a critical role in determining both quality of life and performance outcomes across various activities, particularly in competitive sports (Pradhan et al., 2024). Athletes’ diurnal variation in physiological, cognitive, and physical performance has a significant influence on their competition (Sabzevari Rad, Mahmoodzadeh Hosseini & Shirvani, 2021). Exploring these variables is crucial for optimizing athletic performance during training sessions and competitions. To enhance training adaptations and reduce injury susceptibility, coaches and athletes can take advantage of these studies by aligning their training schedules with their circadian rhythms (Sabzevari Rad, Mahmoodzadeh Hosseini & Shirvani, 2021). In this context, a recent systematic review has demonstrated that morning warm-ups are effective in attenuating intraday fluctuations in performance (Kusumoto et al., 2021).

Chronobiological studies have confirmed that these parameters are dependent on the time of day (TOD) among athletes, with better performance in the afternoon compared to the morning (Chtourou et al., 2013b; López-Samanes et al., 2017; Pullinger et al., 2018; Lopes-Silva, Santos & Franchini, 2019). Peak short-term performance varies significantly throughout the day, consistently demonstrating 3–21% better results in the afternoon compared to morning periods (Chtourou et al., 2014). Recent studies have shown that physical performance, including agility, vertical jump, and repeated sprint performance, is lower in the morning than in the afternoon (Mhenni et al., 2017; Pavlović et al., 2018; Bougrine et al., 2023d). Recently, it has been shown that the 30s cycling exercise (peak power (PP), average power (AP)), the squat jump (SJ) test, and the countermovement jump (CMJ) test were better in the afternoon than in the morning (Bougrine et al., 2022; Bougrine et al., 2023a; Jribi et al., 2024). While some studies (Gholamhasan et al., 2013; Nikolaidis et al., 2018; Unver & Atan, 2015) did not reveal any diurnal variation of explosive effort among athletes, many studies observed differences between morning and afternoon in these parameters among athletes (Reilly et al., 2007; Hammouda et al., 2012; Souissi et al., 2013; Chtourou et al., 2013a; Pallarés et al., 2015; Pavlović et al., 2018; Bougrine et al., 2022; Bougrine et al., 2023d).

This circadian difference is partly explained by the rhythmic variations in resting body temperature, which serve as a “passive warm-up” (Melhim, 1993; Bernard et al., 1997). According to Bougrine et al. (2022), the greater productivity in the afternoon could be explained by several factors related to the circadian rhythm. Indeed, it is at this TOD that body temperature reaches its peak, linked to the individual’s circadian phenotype and their internal biological clock, which is synchronized with their wake-up time. These circadian variations would also influence cognitive performance (Bougrine et al., 2022). Nevertheless, an array of variables, including age, nutritional state, regular training time (Chtourou & Souissi, 2012), training regimen, degree of fitness, the type and intensity of exercise, and lack of sleep, can affect this daily variability in performance (Souissi et al., 2020; Saidi et al., 2021).

While recent data indicate that performance is generally poorer in the early morning and late evening compared to the afternoon period (Ayala et al., 2021), the majority of these studies have only compared two measurements: one at noon and the other in the afternoon. However, we suggest that these time intervals are too large and that there may be performance differences even within the same period of the day, such as between different hours or the afternoon.

Soccer is characterized by a frequent alternation between phases of intense activity, such as sprinting, jumping, shooting, acceleration and deceleration, change of direction, and phases of more moderate activity, such as jogging, walking, or even being stationary (Dugdale et al., 2019; Clemente et al., 2019; Krolo et al., 2020) is not far from these observed TOD effects. Thus, the anaerobic demands of soccer, including agility, sprinting, and power, present themselves as a crucial physiological parameter of performance in soccer players during competition and likely reflect similar diurnal variations, which impact performance at different times of the day. Apart from the lack of focus on diurnal fluctuations, the majority of studies investigating the impact of TOD on sports-related physical performance have not explored the potential effects of this variable in young adolescent athletes. Interestingly, the trend of early specialization in sports seems to be on the rise among young athletes (Farrey, 2008; Myer et al., 2015; Bougrine et al., 2024a), with the pressure to focus on one sport coming from coaches, parents, and peers (Gould, 2009). The performance advantages observed in adults may not apply to younger individuals, underscoring the need for further research in the youth athlete demographic (Farrey, 2008; Upenieks, Ryan & Knoester, 2024). Since the guidelines for adults cannot be applied to adolescent and young athletes, specialized guidance on training timing is needed to help these athletes achieve their performance goals and minimize injury risk.

Although coaches and scouts primarily assess players during competitive matches (soccer’s most dynamic-ecological setting) (Roberts et al., 2021; Abate Daga et al., 2023), significant research gaps remain regarding how diurnal variations, habitual training schedules (Chtourou & Souissi, 2012), and dynamic-ecological methods (Sannicandro et al., 2024) collectively influence youth players’ motor performance, factors that directly impact talent identification. While recent studies confirm both the physical benefits of dynamic-ecological training and afternoon peaks in motor performance (Sannicandro et al., 2024), the critical interaction between circadian rhythms and technical-tactical development remains poorly understood in youth soccer. Therefore, there is a lack of studies investigating the diurnal variability of performance indicators among several days of the week, particularly in the young demographic. Indeed, performance variations may differ depending on whether we compare several time slots around noon or in the afternoon. However, to our knowledge, no study has yet examined the effect of TOD on different aspects of maximal short-term physical performance in adolescent (U15) soccer players by comparing four measurements taken at different times of day, specifically between morning and afternoon.

The objective of this study was, therefore, to evaluate the impact of TOD on maximal short-term physical performance in these young soccer players by comparing four time slots. The selected times capture both established circadian performance variations (morning nadirs (08:00 h) and evening anaerobic performance peaks (18:00 h) and understudied transitional periods (morning-midday progression at 11:00 h and post-lunch dip at 15:00 h). This design addresses key research gaps in adolescent athletes’ diurnal variations while aligning with their actual academic and training schedules for practical relevance.

Based on existing literature (Bougrine et al., 2022; Bougrine et al., 2023a; Bougrine et al., 2024b), we hypothesize that the young soccer players’ maximal short-term physical performance will be significantly affected by the TOD, with better outcomes in the afternoon time slots (18:00 h) compared to the other time slots (08:00 h, 11:00 h, and 18:00 h).

Materials and Methods

Participants

The minimum required sample size was calculated using the software G*power (version 3.1.9.6; Kiel University, Kiel, Germany) (Faul et al., 2007), following the recommendations provided by Beck (2013), with the F test family (repeated measures, within factors) and four conditions. The alpha level was set at 0.05, and the desired statistical power was 0.80. Based on Martín-López et al. (2022) study and the discussion between authors, effect sizes were assessed to be Cohen’s d = 0.7, equivalent to Cohen’s f = 0.35.

Based on this consensus, it was determined that to reduce the possibility of a Type II statistical error in the study, a minimum cohort of 19 athletes would be required. Thirty-five surveys have been examined to reduce dropout rates, and 22 athletes have been identified as meeting our inclusion criteria. Three participants nevertheless dropped out of the research. As a result, our analysis included nineteen athletes who met the research criteria and completed all experimental sessions for our study.

Nineteen young male soccer players (age: 14.58 ± 0.77 years; height: 171.3 ± 8.6 cm; body mass: 57 ± 12.8 kg; BMI: 19.2 ± 2.7 kg/m2) participated in this investigation. All of these players belonged to the same soccer team, the Professional Soccer Academy, and had more than two years of soccer experience (3 ± 1.41 years). Athletes engaged in training at least twice a week, with an average of 3.11 ± 0.74 sessions per week (Table 1).

Table 1 Baseline characteristics of adolescent soccer players participating in the study (n = 19).

Values are presented as mean ± standard deviation (SD), along with minimum and maximum values where applicable. Variables include age, body mass, height, body mass index (BMI), Morningness–Eveningness Questionnaire (MEQ) score, Pittsburgh Sleep Quality Index (PSQI) score, average sleep duration, years of sports experience, frequency of weekly training sessions, usual time of day (TOD) for training, and average duration of training sessions. MEQ scores range from 16 to 86; PSQI scores range from 0 to 21.

	Minimum	Maximum	Mean	
Age (years)	13	15	14.58 ± 0.77	
Body mass (kg)	40.7	80.2	57 ± 12.8	
Height (m)	1.58	1.83	1.71  ± 0.09	
BMI (kg/m 2 )	16.1	25.6	19.2 ± 2.7	
MEQ score (au)
(ranging from 16–86)	43	56	50.47 ± 4.09	
PSQI score (au)
(ranging from 0 to 21)	1	6	2.89 ± 1.45	
Sleep duration (h)	7	10	8.21 ± 0.98	
Experience (years)	2	6	3 ± 1.41	
Training sessions frequency/week	2	5	3.11 ± 0.74	
Usual Training TOD (h)	17:00	17:00	17:00±0	
Usual training session duration (min)	75	75	75±0	
Notes.

MEQ, morningness-eveningness questionnaire of Horne & Ostberg (1976); TOD, Time of day. The minimum, maximum, mean, and standard deviation values of the participants’ characteristics are shown in the table.

The inclusion criteria were as follows: (a) participants were between 13 and 15 years old. (c) Having prior soccer training experience of at least two years. (c) Having a frequency of at least two training sessions per week.

The following were the exclusion criteria: (a) the identification of any disease or illness that might impair performance on different tests and/or the use of any drugs for any long-term medical condition; (b) a habitual consumption of caffeine above 100 mg per day; (c) the use of any medication (including stimulants, narcotics, or psychotropic medicines), dietary supplements, or any restrictive diet that may have affected hormone levels within the previous three months); (d) sleep issues and/or the consumption of alcohol or tobacco; (e) displaying and extreme chronotype.

To minimize the impact of circadian typology on our investigation results, participants displaying an extreme chronotypes (morning or evening chronotypes) were excluded from our study. Only participants with an “Intermediate” chronotype were included in this study, as assessed by their responses to the Horne & Ostberg (1976) self-assessment questionnaire. This questionnaire consists of 19 items that examine sleep and activity preferences, measured on a 4- to 5-point Likert scale, with a total range of 16 to 86. Only athletes with scores between 42 and 58, referring to intermediate chronotype, were included in our study. Additionally, all athletes had a normal sleep duration of 7.9 ± 0.9 h and a Pittsburgh Sleep Quality Index (PSQI) score of 2.89 ± 1.45 in the month preceding the experimental procedure, as assessed by the Arabic version of the PSQI (Suleiman et al., 2010). Indeed, a PSQI score of 8 or more is considered predictive of inadequate sleep. A valid semi-quantitative self-report caffeine consumption questionnaire was used to determine that all enrolled participants consumed a small amount of caffeine (41.05 ± 12.65 mg/d) (Bühler, Lachenmeier & Winkler, 2014).

The protocol of this study complied with Helsinki’s declaration for human experimentation, the ethical and procedural requirements for Human Chronobiology research (Portaluppi, Smolensky & Touitou, 2010). It was approved by the local Ethics Committee of the High Institute of Sport and Physical Education of Kef, University of Jendouba, Jendouba, Tunisia (CPP: 06/2023).

Since the participants were minors, the parents or legal guardians of all athletes provided written informed consent before their children could participate, and verbal consent was obtained from the participants. This process was carried out after an in-depth explanation of the study’s methodology and a consideration of its possible risks and benefits.

Experimental procedure

The research took place in 2024, specifically from February to March. Testing sessions were conducted under uniform conditions in their usual outdoor training facility. The average temperature and relative humidity during the four experimental sessions of the study were about 24 °C and 55%.

Once the participants satisfied all inclusion criteria and agreed to sign the informed parental consent form, they were instructed to refrain from all types of energy drinks, as well as anti-inflammatory and antioxidant substances, until the completion of the experiment. In the 24 h leading up to each trial, participants were asked to follow these guidelines: (a) maintain their usual training regimen and avoid intense physical activities; (b) eat and replicate similar meals; (c) steer clear of stimulants and energy substances within the day before the trial. Body mass was measured in the morning while participants were fasting, using an electronic balance (Tanita, Tokyo, Japan).

Following two familiarization sessions before the tests to minimize learning effects during the experiment and ensure high-quality results, the young athletes participated in the experimental field at four distinct times, with a minimum 48-hour rest interval between each pair of experimental sessions. In random order, the first session occurred at 08:00 h, the second at 11:00 h, the third at 15:00 h, and the last one at 18:00 h (Fig. 1). An online randomization tool (https://randomizer.org/, accessed 10 February 2024) was used for randomization. The selected testing times (08:00 h, 11:00 h, 15:00 h, and 18:00 h) were strategically chosen to reflect both circadian physiology and real-world training contexts for adolescent athletes. Morning (08:00 h) (Bougrine et al., 2022; Bougrine et al., 2023a; Bougrine et al., 2023d) and evening (18:00 h) (Sedliak et al., 2008; Souissi, Souissi & Chtourou, 2019) timepoints were selected based on established evidence of anaerobic performance nadirs and peaks, respectively (Chtourou & Souissi, 2012; Dergaa et al., 2019; Souissi et al., 2022). The inclusion of 11:00 h (Sedliak et al., 2008; Souissi, Souissi & Chtourou, 2019) and 15:00 h (Facer-Childs, Boiling & Balanos, 2018; Souissi, Souissi & Chtourou, 2019; Renziehausen et al., 2023) addresses critical gaps by capturing transitional periods (morning-midday progression and post-lunch dip, respectively) that are ecologically relevant but understudied in chronobiology research. The selected timepoints correspond to the actual academic schedules, training sessions, and competition times of young athletes, ensuring practical relevance while enabling novel analysis of diurnal performance patterns beyond standard morning-evening comparisons.

Figure 1 Overview of the experimental protocol and testing timeline.

Schematic representation of the study design. Assessments included oral temperature (OT), medicine ball throw test (MBT), 5-meter sprint test (5m-T), 20-meter sprint test (20m-T), Illinois Agility Test (IAT), and Illinois Agility Test with ball dribbling (IAT-B). All time points are expressed in local time (GMT +1 h).

In each experimental session, the young athletes began by taking their oral body temperature. Afterward, they performed a series of tests following a 10-min warm-up of low intensity, which consisted of 5 min of running at 10 km/h and 5 min of dynamic stretching, while wearing soccer cleats. The assessments included in the same order an upper body power test (MBT), 5-m and 20-m sprint tests, and an Illinois Agility Test (IAT). Lastly, a specific Illinois soccer test involving a ball was conducted. A 5-min recovery period was allowed between each two tests (Fig. 1).

Oral temperature

The resting oral temperatures were measured using a calibrated digital clinical thermometer (Omron, Paris, France; accuracy ± 0.05 °C) inserted sublingually for at least 3 min after 10 min of rest while seated.

The two kg medicine ball throw test

Medicine ball throw is the most widely known and indirect test used to evaluate the power of the upper limbs in team sports (Leite et al., 2012; Manske & Reiman, 2013). The players throw the medicine ball as vigorously, far, and straight forward as they can, while keeping their back flush against the wall, and their elbows in towards their sides during the push maneuver. Three maximal throws for distance were performed using a measuring tape, and measurements were recorded in meters from the wall to where the medicine ball landed (Decleve et al., 2020). The best throw was retained for the analysis. With a recovery period of ten seconds between three repetitions, the best performance was maintained. Researchers provided the same verbal encouragement procedures for all athletes.

Sprint performance (5 m and 20 m)

The 20-m sprint test was administered to assess acceleration and sprint ability. Linear speed was measured using a 20 m sprint with timing gates (Witty; Microgate, Bolzano, Italy) set up at 0, 5, and 20 m to calculate the 0–5 m and 0–20 m intervals. Timing gates were placed at approximately hip height for all players, as previously recommended (Yeadon, Kato & Kerwin, 1999). Players were instructed to initiate the sprint when ready and cover the set distance as fast as possible. The subjects completed three trials of the sprint, with a minimum of 3 min rest between each trial. The best performance from each of the three trials was used for analysis. Moreover, investigators delivered consistent verbal encouragement protocols across all trials and participants.

Illinois Agility Test (IAT)

The Illinois Agility Test incorporates acceleration, deceleration, change of direction, and sprinting. The Illinois Agility Test, which changed direction, was reported to have high reliability and validity for team sports (Raya et al., 2013; Hachana et al., 2013). The length of the course is 10 m, and the width is 5 m. Four cones are used to mark the start, finish, and the two turning points. Four more cones are placed down the center, an equal distance apart (spaced 3.3 m apart). Participants started from a standing position 0.5 m behind the starting line to avoid early activation of the timing gates. The duration of their performance was quantified using timing gates (Witty; Microgate, Bolzano, Italy) positioned at the beginning and end points, and the superior outcome from the two trials was documented. Players were given instructions to maximize their running speed while following the designated course in the specified direction to reach the endpoint. The start was self-initiated by the subjects when they felt ready, without any starting signal. Furthermore, investigators delivered consistent verbal encouragement protocols across all trials and subjects.

Illinois Agility test with ball dribbling speed (IAT-B)

IAT-B was similar to the IAT, and each player had to slalom through the markers and dribble the ball as quickly as possible from the start to the finish gate (Makhlouf et al., 2022). This test aims to control the ball only with the feet and complete it as fast as possible. Athletes began the test 0.5 m behind a timing gate (Witty; Microgate, Bolzano, Italy) that was used to record their time at the start and finish lines (the same line for this test). The investigators maintained a consistent verbal encouragement procedure for all subjects.

Statistical analysis

STATISTICA software (StatSoft) was used to analyze the data collected for this investigation. GraphPad Prism 8 (GraphPad Software, San Diego, CA, United States) was used to generate the figures. The means ± SD (standard deviation) values were calculated for each variable. A normal distribution of all the data was verified by the Shapiro–Wilk test. To examine the impact of TOD, a one-way repeated measures ANOVA with four TOD conditions was used. Tukey’s HSD Post hoc test was used to assess for significant differences between means when appropriate. The effect size statistic (ηp2) was used to determine the magnitude of the difference between age groups. The criteria, as outlined by Cohen (1992), were applied to determine the effect sizes: a minor effect size was defined as 0.01, a moderate effect size as 0.06, and a large effect size as 0.14. The study employed Cohen’s d analysis, a standardized measure of effect size, to assess the magnitude of differences between variables. The variables were categorized as follows (Hopkins, 2002): trivial (d ≤ 0.20), small (0.20 < d ≤ 0.60), moderate (0.60 < d ≤ 1.20), large (1.20 < d ≤ 2.0), very large (2.0 < d ≤ 4.0), and extremely large (d > 4.0). A significant level was considered as a p ≤ 0.05.

Results

Oral temperature

The ANOVA revealed a significant effect of TOD on oral temperature (F(3,54) = 42.41; p < 0.001; ηp2 = 0.7), indicating that oral temperature (OT) significantly increased progressively from morning to late afternoon. Compared to 08:00 h, OT had significantly increased at 15:00 h (2.2%, p < 0.001) and 18:00 h (3.4%, p < 0.001), but no significant difference was detected at 11:00 h (p > 0.05). Furthermore, the post hoc test showed a significant difference at 15:00 h (1,6%, p < 0.001) and 18:00 h (2,8%, p < 0.001) compared to 11:00 h with higher value observed at 18:00 h. A significant difference was also detected between 15:00 h and 18:00 h (1,1%, p < 0.001) slots (Fig. 2).

Figure 2 Diurnal variation in oral temperature across four time points.

Mean ± SD values of oral temperature measured at 08:00 h, 11:00 h, 15:00 h, and 18:00 h. Notes: *, significant difference compared with 08:00h (p < 0.05); +, significant difference compared with 11:00h (p < 0.05); ∞, significant difference compared with 15:00h (p < 0.05).

The medicine ball throw Test

With respect to upper limb power, there was a significant effect of TOD (F (3,54) = 23.22; p < 0.001; ηp2 = 0.56). The analysis of post hoc tests revealed that upper limb power significantly increased when comparing morning performance (08:00 h) with those at 11:00 h (10,9%, p < 0.001), 15:00 h (10,1%, p < 0.001), and 18:00 h (16,3%, p < 0.001). Likewise, no significant difference was observed between 11:00 h and 15:00 h (p > 0.05). When compared to 18:00 h, a significant difference was demonstrated at 11:00 h (4.9%, p < 0.05) and 15:00 h (5.7%, p < 0.01) time slots (Fig. 3).

Figure 3 Diurnal variation in upper-body explosive power measured by the medicine ball throw test.

Mean ± SD values of medicine ball throw distances recorded at 08:00 h, 11:00 h, 15:00 h, and 18:00 h. Notes: *, significant difference compared with 08:00h (p < 0.05); +, significant difference compared with 11:00h (p < 0.05); ∞, significant difference compared with 15:00h (p < 0.05).

5-m sprint test

The ANOVA revealed a significant effect of TOD on acceleration during the 5 m sprint (F(3,54) = 20.68; p < 0.001; ηp2 = 0.53), indicating that the duration of the sprint over this distance significantly improved from 08:00 h to 18:00 h. Thus, the post hoc test showed that acceleration improved significantly from 08:00 h to 11:00 h (−8,9%, p < 0.001), 15:00 h (−9%, p < 0.001) and 18:00 h (−15,2%, p < 0.001). However, the comparison of sprint speed over 5 m between 11:00 h and 15:00 h was not significant (p > 0.05). A lower performance sprint was observed at 11:00 h (−7%, p < 0.01) and 15:00 h (−6.8%, p < 0.01) compared to 18:00 h (Fig. 4).

Figure 4 Diurnal variations in sprint performance over short distances.

Mean ± SD values for the 5-meter sprint test (A) and 20-meter sprint test (B) recorded at 08:00 h, 11:00 h, 15:00 h, and 18:00 h. Notes: *, significant difference compared with 08:00h (p < 0.05); +, significant difference compared with 11:00h (p < 0.05); ∞, significant difference compared with 15:00h (p < 0.05).

20-m sprint test

The variation in TOD revealed significant effects on sprint time (F(3,54) = 22.86; p < 0.001; ηp2 = 0.55), indicating that the duration of the 20-meter sprint decreased significantly from morning to afternoon. Thus, the diurnal variation in the 20-meter sprint showed reductions in run times at 11:00 h (−5,1%, p < 0.001), 15:00 h (−3,8%, p < 0.01) and 18:00 h (−7,8%, p < 0.001) when compared to 08:00 h. While, the difference between 11:00 h and 15:00 h was found to be nonsignificant (p > 0.05) a significant difference between 11:00 h and 18:00 h has been demonstrated (−2,8%, p < 0.05). Moreover, there was a significant difference between the 15:00 h and 18:00 h (−4,2%, p < 0.001) slots (Fig. 4).

Illinois Agility test

There was a significant effect of TOD on agility, as measured by the Illinois test (F (3,54) = 33.88; p < 0.001; ηp2 = 0.65). Pairwise comparisons of performances indicated that results improved at 11:00 h (−2.8%, p < 0.1), 15:00 h (−2.2%, p < 0.001), and 18:00 h (−4.1%, p < 0.001) compared to the 08:00 h performance. Additionally, a significant difference was indicated at 11:00 h (−1,4%, p < 0.05) and 15:00 h (−1.9%, p < 0.001) slots when compared to 18:00 h. However, the comparisons between 11:00 h and 15:00 h non-significant (p > 0.05) (Fig. 5).

Figure 5 Diurnal variations in agility performance with and without ball dribbling.

Mean ± SD values for the Illinois Agility Test (IAT) (A) and the Illinois Agility Test with ball dribbling (IAT-B) (B) recorded at four time points: 08:00 h, 11:00 h, 15:00 h, and 18:00 h. Notes: *, significant difference compared with 08:00h (p < 0.05); +, significant difference compared with 11:00h (p < 0.05); ∞, significant difference compared with 15:00h (p < 0.05).

Illinois Agility Test with ball dribbling speed

The statistical analysis conducted using ANOVA revealed a significant effect of TOD on the technical abilities of young soccer players (F(3,54) = 26.21; p < 0.001; ηp2 = 0.59), indicating that the time spent dribbling the ball significantly improved from 08:00 to 18:00 h (p < 0.001). The differences between 08:00 h and 11:00 h (−4.5%, p < 0.001) and between 15:00 h and 18:00 h (−2.5%, p < 0.05) were significant. However, the comparison between the 11:00 h (−2.4%, p < 0.05) and 15:00 h (−4.4%, p < 0.001) slots, when compared to 18:00 h, was significant. However, the comparisons between 11:00 h and 15:00 h were nonsignificant (p > 0.05) (Fig. 5).

Discussion

The purpose of this study was to determine the effect of four TOD (08:00 h, 11:00 h, 15:00 h, and 18:00 h) on various anaerobic physical performances among adolescent soccer players (U15). The present findings verify our initial hypothesis, demonstrating that the time of day (TOD) significantly impacts the maximal short-term performance of young soccer players, with superior results consistently observed at 18:00 h compared to (08:00 h, 11:00 h, and 15:00 h) sessions. The main findings of this study were that maximal short-term physical performances (agility with and without the ball, sprint ability, and power of the upper limbs) are affected by the TOD with a meaningful improvement in the late afternoon (18:00 h). This improvement in physical performance was associated with a significant progressive increase in oral temperature from morning to afternoon.

In line with the current findings, several investigations have demonstrated that peak anaerobic performances are observed between 16:00 h and 18:00 h (Chtourou et al., 2013a; Pavlović et al., 2018; Bougrine et al., 2022; Bougrine et al., 2023b), compared to the morning. In addition, our outcomes align with a recent meta-analysis (Ravindrakumar et al., 2022) that revealed late afternoon and early evening (between 16:00 h and 19:30 h) to be the most favorable TOD for short-term maximal physical performance.

An afternoon improvement in repeated sprints (Mhenni et al., 2021; Bougrine et al., 2022; Bougrine et al., 2023a), jumping (Bougrine et al., 2022; Bougrine et al., 2023a), and agility performances (Mhenni et al., 2021; Bougrine et al., 2022) was observed in team ball players. Concerning the interaction between TOD and sprint performance, a recent systematic review conducted by Ravindrakumar et al. (2022) revealed that singular sprints tended to perform much better in the afternoon than in the morning, with 5 m and 20 m overground running sprint timings decreasing by 10.9% and 10.8%, respectively. Mhenni et al. (2017) revealed that handgrip strength, ball throwing velocity, the modified T-test, and the repeated sprint and jump test (average repeated jump performance) were better in the evening than in the morning. On the other hand, they did not find a significant difference between the best jump and the percentage of improvement (Mhenni et al., 2017). Performances of the 5m shuttle run test (i.e., total distance, maximum distance, and fatigue index) increased at 17:00 h compared to 07:00 h, 09:00 h, 11:00 h, 13:00 h, and 15:00 h (Souissi, Souissi & Chtourou, 2019).

Regarding the medicine ball throws, our results are in line with those of Jarraya, Jarraya & Souissi (2015), which showed that the explosive power and speed of the upper limbs were better at 16:00 h than at 09:00 h in children aged 10 years old. Additionally, when comparing jump height and handgrip strength in the late afternoon to those in the morning, a meta-analysis revealed a substantial difference in power output (Knaier et al., 2019). However, Oueslati et al. (2024) showed no diurnal variation in this performance among both male and female schoolchildren.

Although the precise mechanism of the evening effectiveness is not fully understood, the most widely accepted hypothesis proposes that the variables involving body temperature (Serin & Acar Tek, 2019; Sabzevari Rad, Mahmoodzadeh Hosseini & Shirvani, 2021), physiological, psychological, and metabolic cycles (Bellastella et al., 2019; Aoyama & Shibata, 2020) reach their highest in the afternoon. According to Racinais & Oksa (2010), there is a linear and positive correlation between muscle temperature and performance, with a 1 °C increase in temperature converting into a 2–5% increase in performance. Recently Ayala et al. (2021), revealed that body temperature has a circadian rhythm, peaking in the latter afternoon (16:30–18:30 h), when physical performance reaches its maximum (i.e., agility, speed, power, and distance covered) and indicated that this slot is the most appropriate TOD for several aspects of physical activity. Similarly, several studies (Baccouch et al., 2015; Bougrine et al., 2022) found that core temperature was increased from morning to late afternoon between 15:00 h and 18:00 h. Oral temperature observed during the current study indicated a significant effect of TOD, with higher values recorded at 18:00 h. Higher afternoon core temperatures have been linked to increased muscle glycogenolysis, glycolysis, and high-energy phosphate degradation during exercise (Febbraio et al., 1996), as well as increased action potential conduction velocity (Shephard, 1984). In this regard, the body temperature has been estimated to be 0.9% warmer in the afternoon (Serin & Acar Tek, 2019). Furthermore, the increased involvement of both anaerobic (Chtourou et al., 2012) and aerobic (Souissi et al., 2008) metabolic pathways in energy synthesis during the afternoon may account for the improvement in performance from morning to afternoon. The rise in muscle strength, power, and speed during the day, which are components of repeated agility quality (Chtourou et al., 2013a; Facer-Childs & Brandstaetter, 2015), may account for this. This performance increase during the afternoon can be attributed to improved muscle function (Sabzevari Rad, Mahmoodzadeh Hosseini & Shirvani, 2021), increased hormone levels (Bellastella et al., 2019), and enhanced cognitive functions, such as reaction times (Rosa et al., 2021; Bougrine et al., 2024c).

On the other hand, physiological and cognitive variables that fluctuate with the TOD might represent the cause of the morning (08:00 h and 11:00 h) and the start of the afternoon (15:00 h) decline in these anaerobic performances. In the early morning, body temperature remains close to its nocturnal nadir, which has not yet risen to levels that support optimal physiological functioning. As a result, key metabolic processes and neuromuscular activation are attenuated, likely contributing to diminished physical performance during this time of day (Teo, Newton & McGuigan, 2011; Serin & Acar Tek, 2019; Aoyama & Shibata, 2020). Further reducing vigilance and mental alertness in morning exercise is sleep inertia, which is the transitory grogginess that follows awakening and can affect concentration, coordination, and reaction time (Facer-Childs & Brandstaetter, 2015; Serin & Acar Tek, 2019; Bougrine et al., 2022; Bougrine et al., 2023a; Jribi et al., 2024). Since melatonin levels fluctuate throughout the day, this may, in part, explain the decrease in performance during morning sessions compared to the afternoon and could play a significant role in identifying periods of peak performance among athletes (Papantoniou et al., 2014). Higher morning melatonin levels compared to afternoon levels may impede optimal physical performance by causing feelings of exhaustion or decreased attentiveness.

Interestingly, the “post-lunch dip” effect frequently causes performance to decline right after lunch, which can explain in part the decline in performance observed at 15:00 h. Postprandial metabolic activity directs energy toward digestive processes, resulting in transient reductions in arousal states and cognitive performance. This dip is a normal drop in alertness and energy levels (Valdez, Reilly & Waterhouse, 2008; Oueslati et al., 2024). When taken together, these physiological and cognitive factors influence physical performance in the morning and immediately after lunch, resulting in diurnal variations in performance throughout the day.

Contrary to our results, some studies (Unver & Atan, 2015; Nikolaidis et al., 2018; Söğüt, Ödemiş & Biber, 2024) did not report any daily fluctuations in physical maximal exercise among female athletes. Similarly, several investigations have not observed a significant difference in intraday variation in several aspects of short-term maximal exercise (Mhenni et al., 2017; Nikolaidis et al., 2018; Knaier et al., 2019). Since different chronotypes may have distinct internal biological clocks and motivations from the moment they wake up, variations in chronotype and individual preferences, as well as participant wake-up times, could be responsible for inconsistencies among studies (Facer-Childs & Brandstaetter, 2015). Daytime physical activities are closely linked to an individual’s sleep-wake cycle (Van Dongen & Dinges, 2005), which in turn affects attention, response speed, and psychomotor alertness (Rosa et al., 2021). However, a systematic review and meta-analysis did not find a significant effect of exercise timing on metabolic responses to exercise and 24-hour blood level glucose control (Dighriri et al., 2024).

It has been revealed that the TOD affects performance, as there is an optimal performance interval. Still, it cannot remain the same for each individual because it varies based on their circadian typology (Anderson et al., 2018). Likewise, factors that can alter athletes’ circadian rhythms and thus impact their performance level at different TODs include school schedules, sex, and solar day duration (Testu, 1994). In addition to age (adolescent in our study) and sex (male in our study), the results may be influenced by variations in testing procedures, specific sports requirements, training status, and regular training schedules (Chtourou & Souissi, 2012). The current findings indicate that athletes achieve peak performance when conducting anaerobic training during late afternoon hours. These evidence-based strategies enable practitioners to optimize training environments by accounting for biological factors affecting maximal short-term performance in developing athletes. Consistently scheduling high-intensity sessions in the late afternoon proves especially valuable during critical pre-competition preparation periods.

Limitations

This study has several limitations that should be acknowledged. First, while the findings offer valuable insights into diurnal performance variation, they are specific to young male athletes. They may not apply to females, inactive individuals, or other age groups. To address this, we strictly controlled for sex, age, and training status, ensuring internal validity, and recommended that future studies expand demographic diversity. Second, the absence of physiological or biochemical markers (e.g., hormones, blood lactate, or metabolic indicators), which could have provided deeper mechanistic insights into the observed diurnal variations in performance. Future studies should incorporate such measures to elucidate underlying pathways. However, we prioritized ecologically valid field tests (e.g., sprint performance, oral temperature) to align with practical coaching applications, suggesting future research incorporate metabolic and hormonal monitoring. Third, unassessed chronotype and sleep-behavior variability may influence individual responses. To mitigate this, participants maintained their habitual sleep routines before testing, and individuals with extreme chronotypes were excluded through screening. A further limitation is the absence of subjective evaluations in soccer talent identification. While objective assessments effectively measure players’ motor performance and skills, combining them with subjective evaluations helps mitigate biases (Abate Daga et al., 2023). Findings suggest that integrating both methods improves player classification in soccer schools (Abate Daga et al., 2023). We compensated for this by using standardized, validated tests, but we advocate for integrated assessment approaches in applied settings. Future studies should combine physiological monitoring, chronotype stratification, and mixed-method evaluations further to elucidate diurnal performance mechanisms and their practical implications.

Practical Applications

For adolescent male soccer players, training schedules should be strategically aligned with peak performance times. Coaches should prioritize high-intensity sessions (anaerobic performances) and performance testing in the late afternoon (∼18:00 h) when both oral temperature and physical performance are at their peak. This timing ensures valid performance comparisons and accurate progress tracking while optimizing training adaptations. For unavoidable morning or early afternoon matches, implement extended dynamic warm-ups, motivational music protocols (Bougrine et al., 2025), and targeted supplementation (Bougrine et al., 2024b) to mitigate circadian performance dips. Instead, reserve these slots for tactical, technical, or recovery-based training. Practical monitoring tools, such as oral temperature checks, can help identify individual readiness for maximal efforts. While this study focused on intermediate chronotypes, coaches should consider assessing players’ chronotypes (morning, intermediate, or evening types) and adjust their training schedules accordingly to further optimize performance. By aligning training demands with these diurnal performance trends, practitioners can enhance both immediate output and long-term athlete development. For academies with limited evening field access, reserve late-afternoon slots for advanced groups while scheduling foundational technical work for younger players in morning hours when performance demands are lower. Coaches should combine standardized subjective evaluations (technical skills, decision-making) with objective performance metrics in a weighted scoring system to enhance talent identification and informed decisions regarding playing time. This integrated approach reduces selection bias while maintaining the sport’s human element (Abate Daga et al., 2024).

Conclusion

The findings of this study indicated that the TOD significantly affected different aspects of maximal short-term physical performance. Notably, oral temperature and performance in MDB, 5m sprint, 20 m sprint, IAT, and IAT-B tests revealed progressive increase along the day, with the highest levels observed in the late afternoon (18:00 h). Therefore, from a practical perspective, adolescent male soccer players with an intermediate chronotype may achieve their optimal short-term maximal physical performance around 18:00 h. Nonetheless, there is a support to schedule training sessions in the late afternoon, rather than in the morning or early afternoon, as the current study highlights the lower performance levels observed at those times (08:00 h, 11:00 h,15:00 h). In practice, athletes should favor a late afternoon slot to train, as this allows them to achieve the best level of performance.

Supplemental Information

Supplemental Information 1 Raw Data

Athletes: Participant ID. Age: Age of the participants in years. Height: Height of the participants in meters. Weight: Body weight of the participants in kilograms. BMI: Body Mass Index calculated as weight (kg)/[height (m)]2. Experience (year): Years of experience in the respective sport. Training Timing: Usual time of day for training sessions (HH:MM:SS format). Sessions/Week: Number of training sessions per week.Session Duration: Average duration of each training session (HH:MM:SS format). MEQ Score (au): Morningness-Eveningness Questionnaire score in arbitrary units. PSQI Score (au): Pittsburgh Sleep Quality Index score in arbitrary units.Sleep Duration (h): Average sleep duration in hours. Caffeine (mg): Daily caffeine consumption in milligrams. Abbreviations: BMI: Body Mass Index MEQ: Morningness-Eveningness Questionnaire PSQI: Pittsburgh Sleep Quality Index

Supplemental Information 2 Raw data in French

Athletes: Participant ID. Age: Age of the participants in years. Height: Height of the participants in meters. Weight: Body weight of the participants in kilograms. BMI: Body Mass Index calculated as weight (kg) / [height (m)]2. Experience (year): Years of experience in the respective sport.Training Timing: Usual time of day for training sessions (HH:MM:SS format). Sessions/Week: Number of training sessions per week. Session Duration: Average duration of each training session (HH:MM:SS format). MEQ Score (au): Morningness-Eveningness Questionnaire score in arbitrary units. PSQI Score (au): Pittsburgh Sleep Quality Index score in arbitrary units. Sleep Duration (h): Average sleep duration in hours. Caffeine (mg): Daily caffeine consumption in milligrams. Abbreviations: BMI: Body Mass Index MEQ: Morningness-Eveningness QuestionnairePSQI: Pittsburgh Sleep Quality Index

The author would like to thank the youth players who volunteered to participate in this study.

Additional Information and Declarations

Competing Interests

Author Contributions

Human Ethics

Data Availability

The authors declare there are no competing interests.

Houda Bougrine conceived and designed the experiments, performed the experiments, analyzed the data, prepared figures and/or tables, authored or reviewed drafts of the article, and approved the final draft.

Nidhal Nasser performed the experiments, prepared figures and/or tables, and approved the final draft.

Imed Gandouzi analyzed the data, prepared figures and/or tables, authored or reviewed drafts of the article, and approved the final draft.

Halil ibrahim Ceylan analyzed the data, prepared figures and/or tables, authored or reviewed drafts of the article, and approved the final draft.

Majdi Bouazizi analyzed the data, prepared figures and/or tables, and approved the final draft.

Thierry Paillard performed the experiments, analyzed the data, prepared figures and/or tables, and approved the final draft.

Ismail Dergaa analyzed the data, prepared figures and/or tables, and approved the final draft.

Valentina Stefanica conceived and designed the experiments, prepared figures and/or tables, authored or reviewed drafts of the article, and approved the final draft.

Abderraouf Ben Abderrahman conceived and designed the experiments, performed the experiments, prepared figures and/or tables, and approved the final draft.

The following information was supplied relating to ethical approvals (i.e., approving body and any reference numbers):

CPP: 06/2023 from the local Ethics Committee of the High Institute of Sport and Physical Education of Kef, University of Jendouba, Jendouba,  Tunisia.

The following information was supplied regarding data availability:

The raw measurements are available in the Supplemental Files.

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
