# Peer review of "Effects of four different times of day on various aspects of maximal short-term physical performance in young soccer players"

_PeerJ, doi:10.7717/peerj.20065_

## Round 0.1 · original submission · Major Revisions

Some revisions are needed before acceptance.

**Language Note:** The review process has identified that the English language must be improved. PeerJ can provide language editing services - please contact us at [email protected] for pricing (be sure to provide your manuscript number and title). Alternatively, you should make your own arrangements to improve the language quality and provide details in your response letter. – PeerJ Staff

To avoid confusion, we recommend that you specify in the title that this is association football/soccer.

Reviewer 1 ·

Basic reporting

1a. The article is generally well-written, employing clear and professional English. However, minor grammatical and phrasing issues (e.g., lines 55-60, 97-98, and 418-419) occasionally hinder readability. A thorough proofreading or professional language editing is suggested for overall improvement.

1b. The introduction provides adequate context and background, supported by relevant references. Nevertheless, the rationale for selecting these specific testing times could be articulated more clearly, highlighting why intermediate times (11:00 and 15:00) were chosen explicitly alongside the extremes (08:00 and 18:00).

1c. Include Sannicandro et al. (2024) when discussing the theoretical and methodological foundations of the dynamic–ecological training approach. This recent study provides empirical evidence showing superior physical performance outcomes in youth soccer players trained using this method, which would underline the rationale and relevance of your choice of training methodology. DOI: 10.3390/jfmk9020083

1d. The figures provided are generally clear and informative. However, Figure 1 would benefit from enhanced visual quality and more precise labeling to improve immediate comprehension for readers. Figures 2, 3, and 4 illustrate the results with appropriate labeling and statistical significance markers. To further enhance coherence and readability, uniformity in graphical style and formatting across all figures is suggested.

Experimental design

2a. The research question is clear, relevant, and addresses an evident gap regarding diurnal variations in adolescent football performance. However, the manuscript would benefit from explicitly justifying the selection of specific times (08:00h, 11:00h, 15:00h, 18:00h). It is unclear why intermediate times were selected alongside the more commonly investigated extreme times. Clarifying this decision could enhance the methodological rigor and the justification of the experimental design.

Validity of the findings

3a. Justification for Time Slots Chosen
Although the authors selected four specific times (08:00, 11:00, 15:00, and 18:00), the justification for including intermediate time points (11:00h and 15:00h) in addition to the extremes (morning at 08:00h and evening at 18:00h) could be made clearer. Explicitly stating how these intermediate time points contribute uniquely to the literature would strengthen the methodological rationale.

3b. The authors have utilized appropriate statistical methods (repeated-measures ANOVA), but it would enhance the study's rigor to specify explicitly whether the assumptions of the ANOVA (such as sphericity) were verified, including reporting results of tests like Mauchly's sphericity test.
The power analysis is generally appropriate, with clear details on effect size, alpha, and power. However, the rationale behind choosing the specific effect size of 0.35 ("based on the discussion between authors") needs more explicit justification. Ideally, this should be supported by previous literature or pilot data. Additionally, please specify clearly which statistical test (e.g., ANOVA, t-test) was used in your power calculation to enhance methodological transparency. Finally, given the reduced final sample size (N=19), briefly clarify how you managed potential risks related to this smaller-than-anticipated cohort size, and how you controlled for the increased possibility of Type II errors in your final analyses.

3c. While the study extensively evaluates physical performance, including some additional physiological or biochemical markers (e.g., hormonal levels, blood lactate, or other metabolic indicators) would enhance understanding of the underlying mechanisms driving observed diurnal variations. If not feasible, briefly mentioning this as a limitation or future recommendation would be beneficial.

3d. Integrate Abate Daga et al. (2023) findings when addressing the potential limitations associated with subjective evaluations in more transparent soccer talent identification. Their study highlights the importance of combining subjective perceptions with objective assessments, pointing out the risks associated with exclusively subjective classification methods due to biases like the "Halo Effect." https://doi.org/10.3390/children10050767

3e. Practical Implications:
Reference Abate Daga et al. (2024) to reinforce the argument for combining subjective coach evaluations and objective physical performance measures. This study demonstrated how such integrated assessment methods improve decision-making accuracy concerning talent identification and the distribution of playing time, thus offering valuable insights for practical application and coaching strategies. https://doi.org/10.3390/educsci14121400

3f. The conclusions are clearly presented, directly connected to the original research question, and properly supported by your results. However beneficial not only to explicitly acknowledge the study's limitations (line 455-) but also to explain how these limitations were managed or mitigated during the research process. Providing this additional detail will enhance the transparency and rigor of your conclusions, demonstrating clearly the robustness and reliability of your findings.

Additional comments

The manuscript is well-structured and scientifically sound. However, to achieve clear acceptance, I strongly recommend focusing carefully on the following points during revision:

Power Analysis:
Provide a clear and rigorous justification for selecting your effect size (0.35), ideally referencing existing literature, pilot studies, or preliminary analyses. Explicitly state the statistical test used for your power calculation to ensure transparency.

Limitations:
Acknowledge the limitations of your study and explain explicitly how you managed or controlled these limitations within your experimental design and analysis. This approach will considerably enhance methodological robustness.

Literature Integration:
Incorporate recent and relevant literature into your Introduction and Discussion sections, specifically citing studies that explore the dynamic–ecological approach and the relationship between subjective coach evaluations and objective performance measures. This will strengthen the contextual positioning of your work within current research.

Addressing these aspects thoroughly will enhance the manuscript’s rigor, clarity, and overall quality, supporting its successful acceptance.

Reviewer 2 ·

Basic reporting

Dear authors
I appreciate your work put into conducting this research and preparing the manuscript. The topic is interesting and has practical application value. I have only some minor suggestions to correct, because I found your manuscript of high quality.
One important remark about the graphs - the graph type is not adequate for the data being shown.
Dear authors, in any of the measured variables, you do not know what the result will be between the measurements - for example, you do not know what the result would have been measured at 10:00, so you should use the column graphs instead of the line graphs in every figure.

Table 1 - As I can suspect, there should be BMI instead of IMC
MEQ and PSQI - please add the information about the lowest and highest possible values besides the unit
In some places, there are hyphens inside words - please correct, including the text, legends, and tables
Figure 1 in the PDF I have received as a reviewer is of very low quality - please check and improve
Please divide Figure 2 into separate figures, because there is no need to combine these 2 (oral temperature and medicine ball throw).

Line 190 - should write - Since the participants were minors.

Experimental design

Figure 1 - The experimental design can be misleading. It is stated that 48 h are between the test sessions, which is not true. I suggest being less specific and stating just 2 2-day breaks, or if you want to be precise, you should state 51h, 52h, and 51h of rest, respectively, between 8, 11, 15, and 18 test sessions.
In the methods section, you state that the order of time of the sessions was random, but you do not describe the method of randomisation.
2.4, 2.5, 2.6 sprint performance, IAT, IAT-B - please state clearly that there was no starting signal. Describe the starting position and distance to the starting line. Please also give the readers information if there were any verbal encouragement was provided to the participants.
Line 274 - please use the high index for the partial eta squared, and check through the whole text.

Validity of the findings

The statistical analysis is well performed. The graphs should be corrected as I have described above.

Additional comments

I would recommend separating the limitations section from the discussion section. Finish the discussion section with one strong sentence summarising your discussion. I would also recommend adding a practical application section where the recommendations to coaches and other practitioners should be presented.
I would also suggest that the language correction be made by a professional.

Reviewer 3 ·

Basic reporting

This study investigates the effects of different times of day (TOD) on short-term maximal physical performance among adolescent football players. The manuscript has the merit of providing useful practical applications. However, it presents several language issues that have to be addressed. Moreover, the introduction and discussion sections present redundant and repetitive information.

Specific comments

Abstract
Please summarize statistical findings more concisely. Perhaps the authors may focus on effect direction and practical relevance.

Introduction

I suggest that the authors reduce the text within the introduction section by consolidating the circadian rhythm concepts and focusing earlier on the specific research gap.

Line 52: "...determining their level of success or failure...". Revise it.
Line 55: "the adaptation to these stimuli and their optimal employment"..revise here and throughout the manuscript. Remove hyphens where unnecessary.

Line 64-66: "To maximize the benefits of exercise and minimize injury risk", perhaps better using a formal sentence. An example here, "To enhance training adaptations and reduce injury susceptibility".

Lines 69-73"...maximal short-term performance fluctuates throughout the day, with superior results observed in the afternoon compared to the morning, with a difference...". Please revise it, redundant term (i..e, with)

Lines 77-79:The statement regarding power tests and music appears less relevant. I suggest relocating it to the discussion.

Lines 134-139: Time slots are redundant here. Specify them only within the methods section.

Experimental design

Methods

Be consistent when using the acronym IAT-B or IAT-BALL.
Please justify why you decided on an effect size of 0.35. Is it referenced?
Line 151: Use periods instead of commas for decimals.

What about the test and retest reliability of each physical performance assessment?

Validity of the findings

Results

Lines 284-285: "F(3–54)"do not use hyphen as in this case, rather "F(3,54)".
Line 290: remove a bracket

Lines 305–310: As far as I am concerned, "11:00h (-7%, p < 0.01)" appears confusing. For instance, it should be clarified whether it is toward 18:00h or not.

Discussion

Please state whether the initial hypothesis was verified.
Again, redundant concepts here. Please trim the discussion by focusing on the comparison of current findings to novel studies or unresolved contradictions.

Line 430: “The body uses energy for digesting after a meal…” appears to be very informal phrasing. Revise it.

References

Inconsistent formatting of author initials and journal names (e.g., “et al.,” missing standard italicization in some entries).

Please use consistent formatting per journal guidelines.

---

## Round 0.2 · Minor Revisions

Some minor revisions are needed.

Reviewer 1 ·

Basic reporting

The background section provides a solid and up-to-date literature review, with appropriate citations that justify the rationale for the study.

Experimental design

no comment

Validity of the findings

One minor suggestion is to specify whether sphericity was tested (e.g., Mauchly’s test), and if violated, whether any correction (e.g., Greenhouse-Geisser) was applied—this is particularly relevant for repeated-measures ANOVA with more than two levels.

Additional comments

no comment

Reviewer 3 ·

Basic reporting

The authors put a lot of effort in addressing the points raised by the reviewers. The manuscript now has increased its quality. However, I would suggest further changes to achieve the highest standards as possible.


I would suggest modify the title by replacing "youth" to "young"
line 135: is it logical for the hypothesis to include the 18:00h time slot?
Line 172: remove a comma
line 312: suggested rewording --> "With respect to upper limb power, there was a significant effect of TOD"
line 348: replace comma with full stop with decimals
line 163: perhaps "displaying an extreme..."
line 492: "but we advocate"
line 33: MDT or MBT? Please clarify

Experimental design

no comment

Validity of the findings

no comment

---

## Round 0.3 · accepted · Accept

The authors have successfully addressed all reviewers' comments.

Reviewer 3 ·

Basic reporting

no comment

Experimental design

no comment

Validity of the findings

no comment

Additional comments

no comment